# Short-Term Inhibition of Translation by Cycloheximide Concurrently Affects Mitochondrial Function and Insulin Secretion in Islets from Female Mice

**DOI:** 10.3390/ijms242015464

**Published:** 2023-10-23

**Authors:** Mohammed Alshafei, Torben Schulze, Mai Morsi, Uwe Panten, Ingo Rustenbeck

**Affiliations:** 1Institute of Pharmacology, Toxicology and Clinical Pharmacy, Technische Universität Braunschweig, D38106 Braunschweig, Germany; mohammed.alshafei@tu-bs.de (M.A.); torbenschulze@o2online.de (T.S.); mai.morsi@yahoo.com (M.M.); u.panten@tu-bs.de (U.P.); 2Department of Pharmacology, Faculty of Pharmacy, Assiut University, Assiut 71526, Egypt

**Keywords:** cytosolic calcium concentration, insulin secretion, metabolic amplification, mitochondria, pancreatic islets

## Abstract

Since glucose stimulates protein biosynthesis in beta cells concomitantly with the stimulation of insulin release, the possible interaction of both processes was explored. The protein biosynthesis was inhibited by 10 μM cycloheximide (CHX) 60 min prior to the stimulation of perifused, freshly isolated or 22 h-cultured NMRI mouse islets. CHX reduced the insulinotropic effect of 25 mM glucose or 500 μM tolbutamide in fresh but not in cultured islets. In cultured islets the second phase of glucose stimulation was even enhanced. In fresh and in cultured islets CHX strongly reduced the content of proinsulin, but not of insulin, and moderately diminished the [Ca^2+^]_i_ increase during stimulation. The oxygen consumption rate (OCR) of fresh islets was about 50% higher than that of cultured islets at basal glucose and was significantly increased by glucose but not tolbutamide. In fresh, but not in cultured, islets CHX diminished the glucose-induced OCR increase and changes in the NAD(P)H- and FAD-autofluorescence. It is concluded that short-term CHX exposure interferes with the signal function of the mitochondria, which have different working conditions in fresh and in cultured islets. The interference may not be an off-target effect but may result from the inhibited cytosolic synthesis of mitochondrial proteins.

## 1. Introduction

The pancreatic beta cell measures the availability of glucose, along with other nutrients, in the blood stream and transforms it into rates of insulin secretion. While the extent and kinetics of secretion is modified by neuronal and hormonal input, the conditio sine qua non for the full range of stimulated secretion is the presence of glucose or, more generally speaking, of nutrient secretagogues [1,2]. The acute increase in the glucose concentration generates a biphasic response, consisting of the first phase with a rapid onset followed by a transient decrease after 10–15 min of continuous stimulation. The subsequent slow rise in insulin secretion rate forms the second phase which can last for hours [3,4]. The biphasic pattern, which can be observed in vivo and in vitro, is the hallmark of the functional competence of the endocrine pancreas and apparently necessary for the full efficiency of insulin action [5].

The mechanisms underlying the biphasic pattern are still incompletely understood in spite of intensive research activities in the area. It is generally agreed that the metabolism of glucose leads to the closure of ATP-sensitive K^+^ channels and that the ensuing depolarisation of the plasma membrane is the signal for voltage-dependent Ca^2+^ channels to open. While influx of Ca^2+^ is the indispensable signal for the insulin granules to fuse with the plasma membrane, the depolarisation-induced Ca^2+^ influx alone does not fully explain the kinetics of secretion [6]. The ability of glucose to stimulate insulin secretion beyond the extent established by depolarisation [7,8], has led to the hypothesis of an additional pathway enhancing the secretion without affecting the open probability of the K_ATP_ channel and by consequence without affecting the submembrane Ca^2+^ levels [9,10].

According to a suggestion by Henquin this additional pathway is now mostly named the “amplifying pathway”, whereas the pathway involving the closure of the K_ATP_ channels is named the “triggering pathway” [9]. Like the triggering, which is dependent on the glucose-induced ATP production, the amplifying pathway involves signals generated by the mitochondrial metabolism of glucose [11]. To distinguish this mitochondrially derived signalling from the receptor-mediated enhancement of insulin secretion it is termed “metabolic amplification”.

Metabolic amplification likely involves cataplerosis, i.e., the export of citric acid cycle metabolites [12,13], which places it into competition with another essential function of nutrient secretagogues, namely the stimulation of protein synthesis. The beta cell stimulated by high glucose synthesizes (pre)proinsulin, which makes up ca. 50% of total protein biosynthesis under this condition [14,15]. It was calculated that about 25% of the glucose carbon entering the citric acid cycle via anaplerosis is used for protein biosynthesis [16]. Furthermore, it has been hypothesized that proteins with a short half-life are important to ensure optimal Ca^2+^ effects on exocytosis, [17].

While the inhibition of protein biosynthesis can thus be expected to inhibit insulin secretion, as has repeatedly been shown [17,18,19], it may also be possible that the diminished use of non-essential amino acids for protein biosynthesis increases cataplerotic signalling and, thus, insulin secretion. The effects of cycloheximide, a well-characterized inhibitor of eukaryotic translation [20] were investigated both in freshly isolated and in cultured islets, because we recently observed relevant differences between both preparations in their response to glucose stimulation [21].

## 2. Results

For all of the experiments the same protocol was used: 10 μM cycloheximide (CHX) was present in the perifusion medium of isolated islets containing 5 mM glucose, a substimulatory concentration. After 60 min the glucose concentration was raised to 25 mM and, after a further 60 min of perifusion, decreased back to 5 mM. When static incubations were performed, the same sequence was used. In some of the experiments, 500 μM tolbutamide was added instead of raising the glucose concentration.

The initial set of experiments consisted of the simultaneous measurement of oxygen consumption and insulin secretion. Raising the glucose concentration from 5 to 25 mM led to a nearly continuous increase in the secretion rate of fresh islets (from 4 to 70 pg × min^−1^ × islet^−1^) within 60 min. In the presence of CHX the secretion rate was the same at basal glucose but increased more slowly upon stimulation and reached only 56% of the control value after 60 min (Figure 1A). Cultured islets had a lower secretion rate immediately prior to stimulation, but the amount of insulin released during stimulation with 25 mM glucose was not significantly less than the one released by fresh islets. CHX did not significantly diminish the stimulated secretion of the cultured islets (Figure 1B).

At 5 mM glucose, the oxygen consumption rate (OCR) of freshly isolated islets was about 50% higher than the one of cultured islets (Figure 1, Figure 2 and Figure 3C,D). Remarkably, the presence of CHX led to an initial overshooting OCR of the fresh islets (Figure 1A and Figure 2A). Raising glucose to 25 mM led to significant increases in the OCR by fresh and cultured islets, except for fresh islets in the presence of CHX (Figure 3C). This reflects the higher prestimulatory OCR in the presence of CHX, which was not visible with cultured islets. Or, from a different perspective, CHX diminished the glucose-induced OCR increase in fresh, but not in cultured, islets (Figure 3E).

The stimulation of insulin secretion by 500 μM tolbutamide resulted in a much smaller secretory response (ca. 25%) than the stimulation by 25 mM glucose. Again, CHX significantly reduced the secretion of fresh islets, but not of cultured islets (Figure 2 and Figure 3B). Again, the OCR of fresh islets was significantly higher than the one of cultured islets. Tolbutamide stimulation was practically inefficient to increase the OCR under the control condition but was marginally effective in the presence of CHX (Figure 2 and Figure 3D,F).

The ratio of OCR per secreted insulin was lower during stimulated secretion than during basal secretion; this was true for both stimulation by glucose and stimulation by tolbutamide (Figure 4). While the OCR-secretion ratio decreased throughout the entire phase of glucose stimulation, a plateau was reached early during tolbutamide depolarization (Figure 4A–D). During stimulation, the oxygen consumption per released insulin was higher in CHX-exposed fresh islets than in control fresh islets (Figure 4A,B). This difference was also visible during tolbutamide stimulation of cultured islets, but not during glucose stimulation of cultured islets (Figure 4C,D). Or, from a different perspective, CHX increased the difference between fresh and cultured islets, and more clearly so for glucose stimulation than for tolbutamide stimulation (Figure 4E,F).

Unexpectedly, the glucose-induced insulin secretion showed a continuously ascending kinetic, not the typical biphasic pattern (Figure 1). Presuming that the slow perifusion velocity was the reason [22], the same glucose stimulation was performed with a sevenfold higher perifusion velocity to achieve a square wave-pattern of glucose stimulation. Under this condition a biphasic response pattern with a significant nadir after the first phase resulted (Figure 5A,B). The freshly isolated islets produced a continuously ascending second phase, whereas the second phase of the cultured islets resembled more of an elevated plateau, which remained lower than the peak of the first phase (Figure 5B). Like in the initial experiments CHX did not affect the basal secretion rate but reduced the glucose-stimulated secretion of the fresh islets starting at the first time point of stimulated secretion. The biphasic pattern was transformed into a slightly ascending plateau and the amount of released insulin was diminished to 42%. With cultured islets CHX significantly diminished the peak value of the first phase but thereafter the secretion rate gradually increased and was finally significantly stronger than the control secretion. Of note, the stimulated secretion under the control condition was only 57% of the corresponding value of fresh islets (Figure 5A vs. Figure 5B, insets). After 20 min of wash-out of glucose, the secretion rates of CHX-exposed islets and control islets were again identical, both with fresh and cultured islets. Thus, the lack of effect of CHX prior to the stimulation is not due an insufficient exposure time but reflects a specific action on stimulated secretion.

The question whether the inhibition of insulin secretion by CHX involved changes in the cytosolic Ca^2+^ concentration ([Ca^2+^]_i_) was addressed by measuring the Fura fluorescence ratio of intact perifused islets (Figure 6A–D). In the period prior to stimulation a difference between CHX-exposed and control islets gradually developed when the islets were freshly isolated but not when they were cultured. The increased levels of [Ca^2+^]_i_ during stimulation by glucose or tolbutamide were significantly diminished by CHX, both in fresh and in cultured islets (Figure 6E,F). The onset of the [Ca^2+^]_i_ increase and the wash-out characteristics were closely similar irrespective of the presence of CHX (Figure 6A–D). The closely similar effects in fresh and cultured islets are remarkable, since the latter had significantly lower [Ca^2+^]_i_ values than the fresh islets both prior to and during stimulation (Figure 6E,F).

To investigate the extent to which the protein biosynthesis was inhibited by CHX, a sequence of static incubations was performed to mimic the conditions of islet perifusion. The insulin content, but not the proinsulin content, of fresh islets was significantly higher than the one of cultured islets. Proinsulin made up 3.7% of the insulin content in fresh islets and 4.6% in cultured islets (Figure 7). Within 60 min the presence of CHX reduced the proinsulin content, but not the insulin content, to about 50% in fresh and to about 35% in cultured islets (Figure 7B,D). The additional presence of 25 mM glucose for a further 60 min affected neither the content of insulin nor the diminished content of proinsulin (Figure 7A–D).

The mitochondrial function was assessed by measuring the NAD(P)H- and FAD-autofluorescence and the fluorescence of the indicator of the mitochondrial membrane potential, TMRE (Figure 8 and Figure 9). After a short time lag, the increase in the glucose concentration in the perifusion medium of fresh islets led to a virtually simultaneous increase in the NAD(P)H- and decrease in the FAD-fluorescence (Figure 8A). Wash-out near-exponentially decreased the NAD(P)H- and increased the FAD-fluorescence. Practically the same pattern was observed in the presence of CHX (Figure 8B). Calculation of the NAD(P)H/FAD ratio as a semi-quantitative measure of the reducing equivalents in the matrix space showed that in the presence of CHX the increase in this parameter reached only 53% of the control value (Figure 9A vs. Figure 9B).

In cultured islets the autofluorescence showed essentially the same pattern as in fresh islets (Figure 8C,D); however, the NAD(P)H/FAD ratio was significantly higher during glucose stimulation (Figure 9C,D). In contrast to fresh islets, the presence of CHX did not diminish the glucose-induced increase in the NAD(P)H/FAD ratio (99% of the control value). Similarly, the glucose-induced increase in the mitochondrial membrane potential (TMRE fluorescence) was more marked in cultured islets (Figure 9C,D). It was moderately reduced by CHX (to 78% of the control value), but this was still slightly higher than the control value of fresh islets.

The consequence of the effect of CHX on the level of reducing equivalents was checked by measuring the islet content of ATP and ADP after static incubations designed to mimic the perifusion sequence. This was followed by the presence of the uncoupler CCCP, to establish a minimal ATP value (Figure 10). The ATP content of fresh and cultured islets was the same at 5 mM glucose, after stimulation with 25 mM glucose and after renewed incubation with 5 mM glucose. Only the presence of CCCP resulted in a significantly lower ATP content for the cultured islets. The ADP content of cultured islets, however, was significantly lower after all four incubation conditions (Figure 10C), which resulted in significantly higher ATP/ADP ratios in cultured islets during glucose stimulation and after return to basal glucose (Figure 10E).

After all four incubation conditions the ATP content of CHX-exposed fresh islets was closely similar to the one of control fresh islets (Figure 10B). Even though the ADP contents of CHX-exposed fresh islets were only moderately lower (marginally significant at 25 mM glucose, Figure 10D), the resultant ATP/ADP ratio was significantly higher prior to stimulation, during stimulation (marginally) and after exposure to CCCP (Figure 10F). In cultured islets the presence of CHX virtually abolished the stepwise change in the adenine nucleotide content, and no significant differences existed between cultured islets which were CHX-exposed and their respective controls.

## 3. Discussion

The question which started this investigation was whether nutrient-dependent protein biosynthesis in the beta cell interferes with the mitochondrial signal generation for the metabolic amplification of insulin secretion. Since we recently observed differences between the metabolic amplification in fresh islets and in cultured islets [21], all experiments were performed with freshly isolated NMRI mouse islets and with NMRI mouse islets cultured for one day. We found that the reduction in insulin secretion by the short-term inhibition of translation, known since the early days of beta cell research [18,19], was significant only in freshly isolated islets but not in cultured islets.

The lack of effect on cultured islets concurs with the non-significant increase in insulin secretion observed after treatment of cultured islets with 10 μM emetine, which inhibited translation by 84% [23]. When glucose was applied as a square-wave stimulus (see Figure 5), the same short-term inhibition of translation by CHX significantly enhanced the glucose-induced insulin secretion of cultured islets, beginning in the course of the second phase. These divergent effects confirm that nutrient stimulation produces different responses in fresh and cultured mouse islets [21] and offer the opportunity to narrow down the relevant mechanism(s) of inhibition by comparing the parameters of stimulus–secretion coupling of both islet preparations.

The observation that in the presence of basal glucose CHX also inhibited the tolbutamide-stimulated secretion by fresh but not cultured islets was unexpected, because it had been reported earlier that the sulfonylurea-induced increase in insulin secretion was not affected by puromycin, another inhibitor of translation [24]. However, Garcia-Barrado et al. [17] have observed that in the presence of 15 mM glucose the additional stimulation by tolbutamide or 30 mM KCl was virtually abolished by CHX. Since depolarization as such (by K_ATP_ channel-blocking sulfonylureas or high extracellular KCl) does not increase insulin biosynthesis [15], our present observation argues against the inhibition of insulin biosynthesis as the mechanism underlying the inhibition of secretion, but it does not rule out the inhibition of translation as such.

The low concentration of CHX (10 μM) and the short-term pretreatment (60 min) employed here have been shown earlier to reduce protein biosynthesis by 93% [17]. Since we found that the islet content of proinsulin in fresh and in cultured islets was reduced to a comparable degree, it can be concluded that the inhibition of translation by CHX is equally effective in both preparations and does not explain the observed difference in secretion. Cultured islets had a significantly lower insulin content than freshly isolated islets. Since the glucose concentration in the culture medium was 5 mM to avoid constant stimulation, an increased rate of exocytosis seems unlikely as a cause; rather, granule degradation (e.g., by autophagy, [25]) can be assumed. It is known that the glucose dependence of insulin biosynthesis is left-shifted versus the stimulation of secretion [15], which fits to the observation that only the insulin content, but not the proinsulin content, was diminished at the end of the culture period.

CHX diminished the [Ca^2+^]_i_ increase caused by glucose stimulation, which initially appeared to explain the diminished increase in secretion. However, it did so in cultured islets as well, where the secretion was unimpaired or even enhanced. In principle, the same effect of CHX on [Ca^2+^]_i_ was seen when the islets were stimulated by tolbutamide. This rules out [Ca^2+^]_i_ as the direct mediator of the inhibition of secretion and raises two questions: (i) how is the diminished [Ca^2+^]_i_ increase related to the different pattern of secretion and (ii) is the diminished increase in [Ca^2+^]_i_ an off-target effect of CHX or an indirect consequence of the inhibition of translation? The first question can be answered by considering the secretion in response to the square-wave stimulation, the same as was used for the [Ca^2+^]_i_ measurements. Even though the onset of secretion was not retarded, the overshooting secretion of the first phase was significantly reduced in both preparations. For the remaining stimulation period the [Ca^2+^]_i_ increase is apparently sufficient to support stimulated secretion and to permit enhanced amplification in cultured islets.

Since the measurement of the OCR suggested a diminished glucose-induced increase by CHX in fresh but not cultured islets (see Figure 3E), the mitochondrial energetics were characterized by measuring the autofluorescence and the mitochondrial membrane potential. In fresh islets CHX diminished the glucose-induced changes in the NAD(P)H- and the FAD-autofluorescence, whereas in cultured islets the FAD-decrease was unchanged, and the NAD(P)H increase was even enhanced and accelerated. The concomitant increase in the mitochondrial membrane potential by glucose stimulation was moderately reduced by CHX in both preparations, but since the glucose-induced increase was generally higher in cultured islets, the CHX-exposed cultured islets still showed higher values than fresh control islets. At variance with the report by Garcia-Barrado et al. [17] our conclusion is that CHX affects mitochondrial function as soon as it affects secretion, and it does so with different consequences for fresh and cultured islets.

The difference between fresh and cultured islets was also visible in the adenine nucleotide content. Cultured islets showed a significantly higher glucose-induced increase in the ATP/ADP ratio than fresh islets. It has been proposed that this difference is due to the lower granulation state after overnight culture [26], and this may also apply here since the insulin content of the cultured islets was significantly lower than the one of fresh islets (see above). Of note, the higher ATP/ADP ratio was not due to increased levels of ATP, but rather it was due to lower levels of ADP. This feature may result from the lower granulation state since insulin granules do not only contain high amounts of ATP but even higher amounts of ADP [27]. Furthermore, the consumption of cytosolic ADP by the pyruvate kinase reaction may be a contributing factor [28].

In contrast to our expectations, CHX did not decrease the ATP/ADP ratio in fresh islets. It was even somewhat increased, mainly because of the lower ADP content. Considering the ATP/ADP ratio as a pure triggering signal [9,29], this observation fits to the kinetics of the [Ca^2+^]_i_ increase and the onset of significantly elevated secretion rates which were not retarded by CHX (see Figure 5 for the fast perifusion protocol). The basal secretion rates were not affected by CHX; this is confirmed by the return to prestimulatory levels after wash-out of the stimuli in spite of the much longer CHX exposure. Thus, the inhibition of insulin secretion by short-term exposure to CHX may be described as a reduced amplification with unimpaired triggering.

This description may seem improbable in view of the inhibition of the insulinotropic effect of tolbutamide, which, again, was significant in fresh, but not in cultured, islets. While a stimulation of translation by sulfonylureas may occur as a secondary effect after prolonged exposure [30] the acute stimulation of secretion by tolbutamide is not accompanied by increased translation [31]. However, it has to be emphasized that the stimulation of insulin secretion by sulfonylureas is not independent of the glucose metabolism. In the absence of glucose, the insulinotropic effect of a maximally effective sulfonylurea concentration is just achieving significance [32] and can be abolished by the inhibition of oxidative phosphorylation [33]. So, the interference with mitochondrial metabolism by CHX appears to be compatible with the decreased insulinotropic effect of tolbutamide in fresh islets.

The mitochondrial metabolism of fresh and cultured islets differed not only by the level of reducing equivalents, but also by the OCR. The OCR of fresh islets was about 50% higher than the OCR of cultured islets (see Figure 1 and Figure 2). The OCR of fresh islets in the presence of 5 mM glucose was similar to the one of fresh islets without glucose [34], which is probably due to the consumption of endogenous nutrients in the latter case [35]. So, the lower OCR in the cultured islets may not result from a limiting role of nutrients but from a higher mitochondrial membrane potential. A more marked increase in the mitochondrial membrane potential was clearly apparent during glucose stimulation of the cultured islets. The higher level of reducing equivalents in these islets supports this interpretation. Since the ATP content of fresh and cultured control islets was not significantly different, the mitochondria in fresh islets may be less tightly coupled than those in cultured islets.

Even though off-target effects of CHX cannot be ruled out, our interpretation is that all effects reported here result from the inhibition of cytosolic translation by CHX. There are at least two different consequences to consider: First, the block of translation itself, whereby the expenditure of energy is diminished with potentially immediate consequences. Second, the lack of the proteins which are normally produced by cytosolic translation, a process which will require some more time before functional consequences appear. With respect to the changed mitochondrial function by CHX it is relevant that the vast majority of the mitochondrial proteins is produced by cytosolic translation in a coordinated way with intramitochondrial protein biosynthesis [36,37]. It may well be that the hypothetical amplification-mediating protein with a short half-life [17] is among these proteins. The suspected competition between protein biosynthesis and amplification signaling by cataplerosis, which started this investigation, may be the mechanism underlying the increased secretion rates of cultured islets during the second phase. Both of these hypotheses deserve further investigation. Preliminary results obtained with inhibitors of mitochondrial translation confirm the fast onset of altered insulin secretion. The renewal of mitochondrial proteins may be a relevant factor in the glucose-sensing role of the beta cell mitochondria [38].

## 4. Materials and Methods

*Chemicals.* Fura-2 LeakRes (AM) was obtained from TEF-Labs (Austin, TX, USA) and TMRE from AnaSpec (Fremont, CA, USA). Cycloheximide, tolbutamide, the ATP measurement kit, pyruvate kinase, collagenase P and the cell culture medium RPMI 1640 (without glucose) were from Sigma-Aldrich (Taufkirchen, Germany). Fetal calf serum (FCS Gold ADD) was obtained from Bio & Sell (Nürnberg-Feucht, Germany), and bovine serum albumin (BSA, fraction V) and all other reagents of analytical grade were obtained from E. Merck (Darmstadt, Germany).

*Tissue and tissue culture.* Islets were isolated from the pancreas of female NMRI mice (12–14 weeks old, fed ad libitum) by collagenase injection into the bile duct and hand-picked under a stereomicroscope. Care was taken to limit the time between the onset of digestion and beginning the perifusion of freshly isolated islets to 45 min. For a direct comparison between freshly isolated islets and cultured islets, the same batch of islets was split in two. Islets were cultured in cell culture medium RPMI 1640 with 10% FCS in a humidified atmosphere of 95% air and 5% CO_2_ at 37 °C. The glucose concentration was 5 mM, and the culture duration was 22 ± 1 h. Animal care in the central facility of the Technische Universität Braunschweig (internal ID: AZ §4 (08.01) TSB TU BS) is supervised by the regional authority (LAVES, Lower Saxony, Germany) and conforms to the current EU regulations.

*Measurement of insulin secretion and insulin content.* Batches of 50 islets were introduced into a purpose-made perifusion chamber (37 °C) and perifused at 0.9 mL/min with a HEPES-buffered Krebs–Ringer medium (KR-medium), which was saturated with 95% O_2_ and 5% CO_2_ and contained the following (mM): NaCl 118.5, KCl 4.7, CaCl_2_ 2.5, KH_2_PO_4_ 1.2, MgSO_4_ 1.2, NaHCO_3_ 20, HEPES 10 and BSA 0.2% *w*/*v*. The insulin content in the fractionated efflux was determined by ELISA according to the manufacturer’s protocol (Mercodia, Uppsala, Sweden). The islet content of insulin and proinsulin was measured by sonicating a group of 20 islets in an ice-cooled microtube for 20 s and measuring the insulin concentration after appropriate dilution with the zero buffer of the respective ELISA set (Mercodia).

*Simultaneous measurement of islet oxygen consumption and insulin secretion.* The inflow and the outflow of the perifusion chamber were conducted through miniaturized oxygen sensors (Pst3 sensors and Fibox4 m, PreSens, Regensburg, Germany). The perifusion media were constantly equilibrated with a gas mixture containing 21 vol% oxygen. Before passing the first sensor the media passed an “artificial lung” equilibrator [39] to generate constant oxygen levels. The difference between the sensors divided by the number of islets gave the oxygen consumption rate (OCR). To achieve stable recordings, a higher number of islets (150) and a slower pump rate (125 µL/min) were necessary for the secretion measurements [34]. To describe the relation between OCR and insulin secretion, the fractionated efflux was collected, and the insulin content was measured by ELISA. The kinetics of secretion were corrected for the lag time between the perifusion chamber and the fraction collector, and the OCR kinetics were corrected for the lag time between the upstream and the downstream sensor.

*Measurement of the NAD(P)H- and FAD-autofluorescence and of the TMRE-fluorescence.* The fluorescence of NAD(P)H (= NADH + NADPH), FAD [21,40], and of the indicator of the mitochondrial membrane potential, TMRE [13], were simultaneously recorded. After loading with TMRE (20 nM for 30 min), one islet was placed in a purpose-made perifusion chamber on the stage of an Orthoplan epifluorescence microscope (Leitz/Leica, Wetzlar, Germany) and perifused at 0.2 mL/min. Fluorescence of a subregion of the islet was excited with a 150W xenon arc using the following filter combinations (Omega Optical, Brattleboro, VT, USA): for NAD(P)H, excitation 366 ± 15 nm bandpass, dichroic separation 405 nm, and emission 450 ± 32 nm bandpass; for FAD, excitation 440 ± 21 nm bandpass, dichroic separation 455 nm, and emission 520 ± 20 nm bandpass; and for TMRE, excitation 535 ± 18 nm bandpass, dichroic separation 570 nm, and emission 590 ± 18 nm bandpass. The filter cubes were switched every 2.5 s with an exposure time of 0.1 s. The fluorescence emission was collected by a Zeiss Fluar objective (40×, 1.3 N.A.) and measured by a photon-counting multiplier. For the calculation of the mean values the fluorescence intensities were normalized to 100% at the last prestimulatory time point (60 min) in each experiment. The NAD(P)H- and FAD-traces were then multiplied with the mean intensity values at 60 min to make the different data sets comparable. As a semiquantitative parameter of the level of mitochondrial reducing equivalents the NAD(P)H/FAD ratio was calculated.

*Islet content of adenine nucleotides.* Fifteen islets were statically incubated to mimic typical perifusion conditions. After precipitation of the proteins and extraction of the adenine nucleotides, ATP was determined by use of the luciferase method as previously described [32]. The ADP content of the extract was converted into ATP by the pyruvate kinase reaction, with the difference between both measurements yielding the net ADP content. Because of the interindividual variations in the adenine nucleotide contents, the incubations were strictly performed in parallel and comparison between cultured and fresh islets was only made with islets from the same isolation batch.

*Measurement of the cytosolic Ca^2+^ concentration ([Ca^2+^]_i_)*. Freshly isolated or cultured islets were incubated in Krebs–Ringer medium (5 mM glucose) with Fura-2 LeakRes (AM) at a concentration of 2 μM for 45 min at 37 °C. Five islets were then inserted in a temperature-controlled perifusion chamber (35 °C) on the stage of a Zeiss Axiovert 135 microscope equipped with a Zeiss Fluar (10×, 0.5 N.A.) objective. The islets were perifused with a HEPES-buffered Krebs–Ringer medium, which was saturated with 95% O_2_ and 5% CO_2._ The fluorescence of each islet (excitation at 340 or 380 nm, dichroic separation at 410 nm and emission at 510 ± 40 nm) was recorded with a cooled CCD camera (Pursuit, Diagnostics Instruments, Sterling Heights, MI, USA) and evaluated using Visiview software (Visitron, Munich, Germany).

*Statistics.* Results are presented as the mean ± SEM. GraphPad Prism5 software (GraphPad, LaJolla, CA, USA) was used for statistic calculations and non-linear curve fitting (ELISA). If not stated otherwise “*t*-test” refers to the unpaired, two-sided *t*-test and “significant” refers to *p* < 0.05. A marginal significance (0.05 ≤ *p* ≤ 0.07) is indicated by a symbol in parentheses.

## 5. Conclusions

Thus far, the inhibition of insulin secretion by acute inhibition of protein biosynthesis has only been considered with regard to its relevance for insulin synthesis or exocytosis-regulating proteins. The present observations suggest that in the beta cell the mitochondrial function is affected very soon after the translation is inhibited, coinciding with the inhibition of secretion. The different consequences for the insulin secretion of freshly isolated islets vs. that of cultured islets correlate with the different working conditions of the mitochondria in these preparations.

## Figures and Tables

**Figure 1 ijms-24-15464-f001:**
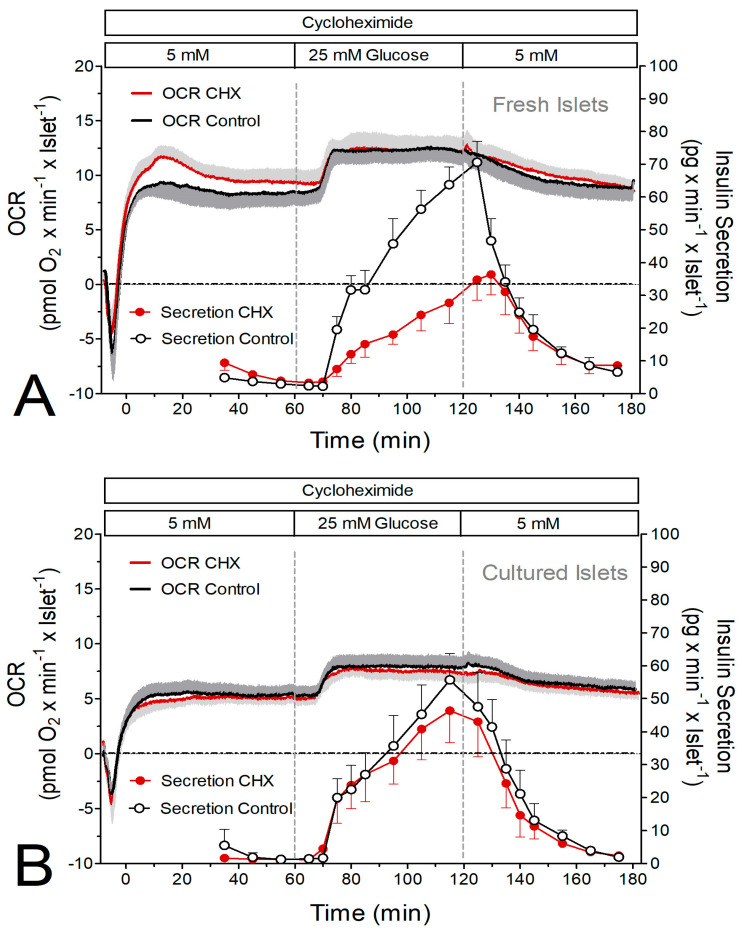
Simultaneous measurement of glucose-induced insulin secretion and oxygen consumption rate (OCR) by perifused islets in the presence and absence of the inhibitor of translation, cycloheximide (CHX). (**A**) Freshly isolated islets were perifused with Krebs–Ringer medium in the presence (red) or absence (black) of 10 μM CHX for 180 min. After 60 min the glucose concentration was raised from 5 to 25 mM and after a further 60 min it was lowered back to 5 mM. (**B**) Twenty-two-hour-cultured islets were perifused with Krebs–Ringer medium in the presence (red) or absence (black) of 10 μM CHX for 180 min. After 60 min the glucose concentration was raised from 5 to 25 mM and after a further 60 min it was lowered back to 5 mM. The traces indicate the oxygen measurements, and the circles indicate the measurement of insulin in the fractionated efflux. Values are means ± SEM of 6 experiments each.

**Figure 2 ijms-24-15464-f002:**
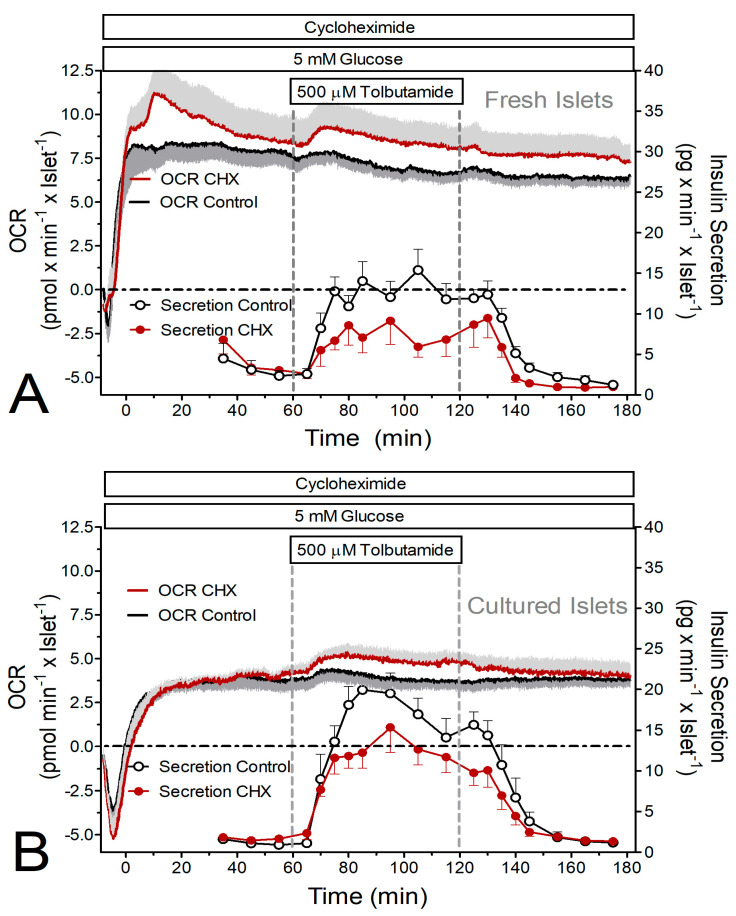
Simultaneous measurement of tolbutamide-induced insulin secretion and oxygen consumption rate (OCR) by perifused islets in the presence and absence of the inhibitor of translation, cycloheximide (CHX). (**A**) Freshly isolated islets were perifused with Krebs–Ringer medium in the presence (red) or absence (black) of 10 μM CHX for 180 min. After 60 min 500 μM tolbutamide was added to the perifusion medium containing 5 mM glucose. After a further 60 min, tolbutamide was washed out. (**B**) Twenty-two-hour-cultured islets were perifused with Krebs–Ringer medium in the presence (red) or absence (black) of 10 μM CHX for 180 min. After 60 min, 500 μM tolbutamide was added to the perifusion medium containing 5 mM glucose. After a further 60 min, tolbutamide was washed out. The traces indicate the oxygen measurements, and the circles indicate the measurement of insulin in the fractionated efflux. Values are means ± SEM of 5 experiments each.

**Figure 3 ijms-24-15464-f003:**
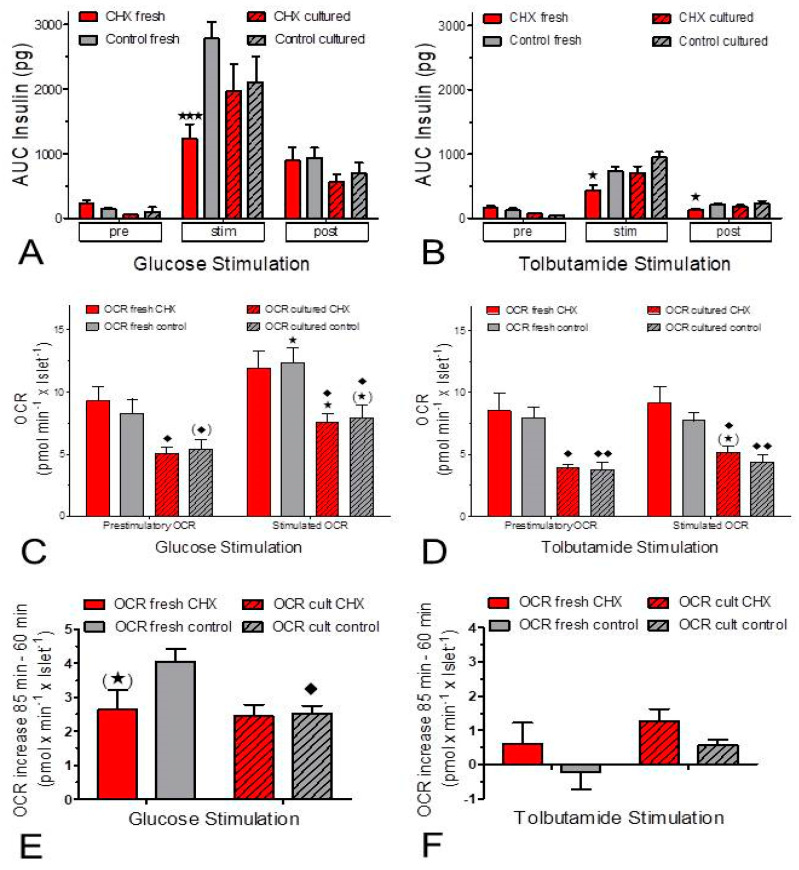
Side-by-side comparison of the insulin secretion (**A**,**B**) and oxygen consumption rate (OCR, **C**–**F**) as derived from Figure 1 and Figure 2. Fresh islets are indicated by open bars and cultured islets by hatched bars; the presence of cycloheximide (CHX) is indicated by red bars, and the control condition is indicated by grey bars. In (**A**,**B**) the asterisks denote significant differences between CHX and control (* *p* ˂ 0.05, *** *p* ˂ 0.001, *t*-test). In (**C**,**D**) the asterisks denote significant differences between the stimulated and prestimulatory OCR (* *p* ˂ 0.05, *t*-test) and the rhombus symbols denote significant differences between fresh islets and cultured islets (^♦^ *p* ˂ 0.05, *t*-test, ^♦♦^ *p* ˂ 0.01, *t*-test). In (**E**,**F**) the OCR increases induced by glucose and by tolbutamide are plotted. The asterisk in parentheses indicates a marginal significant difference (0.05 ≤ *p* ≤ 0.07, paired *t*-test) between CHX and control, the rhombus symbol denotes a significant difference between fresh islets and cultured islets (^♦^ *p* ˂ 0.05, paired *t*-test).

**Figure 4 ijms-24-15464-f004:**
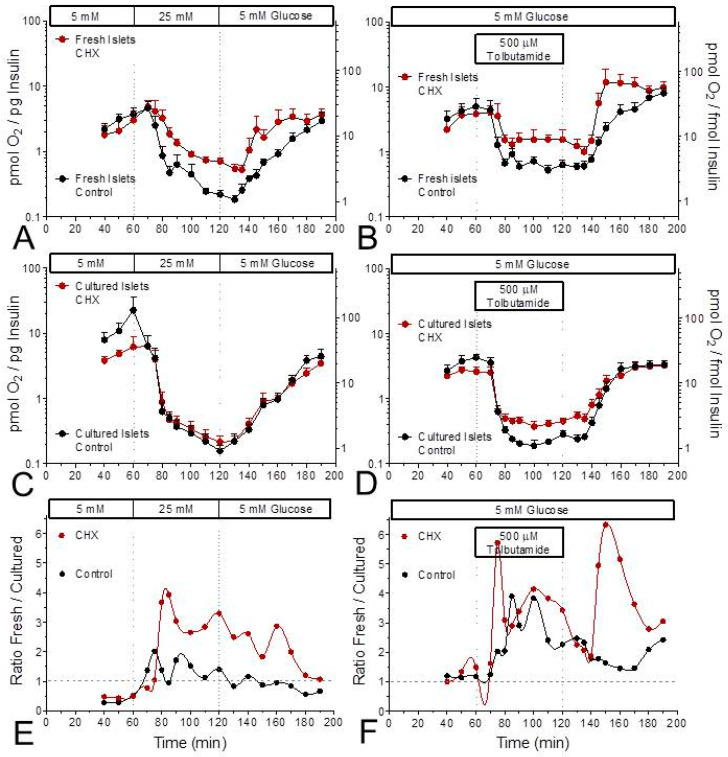
Relation between glucose-induced insulin secretion and oxygen consumption in the presence and absence of the inhibitor of translation, cycloheximide (CHX). Data from Figure 1 and Figure 2 were plotted as the ratio of insulin secretion versus oxygen consumption rate (OCR) for glucose (left column) or tolbutamide stimulation (right column). The data of fresh islets are shown in the upper row (**A**,**B**), those of cultured islets in the middle row (**C**,**D**). In each graph the presence of CHX is indicated by red symbols, the control condition by black symbols. Note the logarithmic scales. The difference between CHX and control during glucose stimulation of fresh islets (**A**) is not visible with cultured islets (**C**). The lower row (**E**,**F**) shows the ratio of fresh islets by cultured islets. CHX continuously elevates the ratio for glucose stimulation but does so only at the start and during wash-out for tolbutamide stimulation.

**Figure 5 ijms-24-15464-f005:**
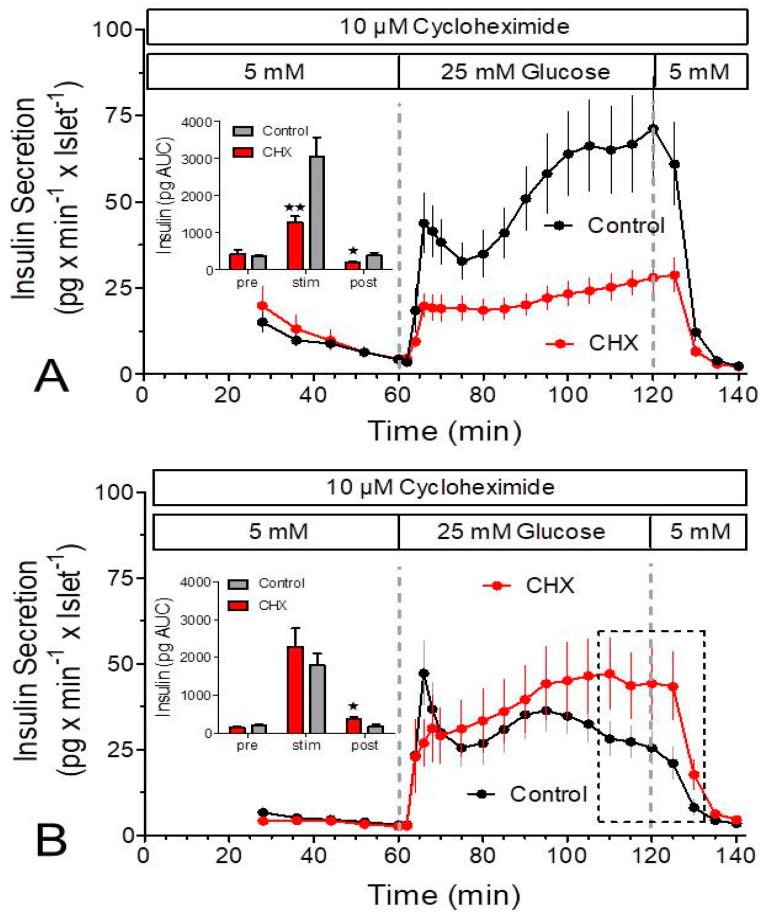
Measurement of insulin secretion in the presence and absence of cycloheximide (CHX) by islets exposed to a square-wave glucose stimulus. Freshly isolated (**A**) or cultured islets (**B**) were perifused at 0.9 mL/min with Krebs–Ringer medium in the presence (red) or absence (black) of 10 μM cycloheximide (CHX) for 140 min. After 60 min the glucose concentration was raised from 5 to 25 mM, and after a further 60 min it was lowered back to 5 mM. Note the clear biphasic pattern both in fresh and in cultured control islets. Asterisks denote significant differences between CHX and control (* *p* ˂ 0.05, ** *p* ˂ 0.01, *t*-test). The dashed rectangle in (**B**) denotes the values where the secretion was significantly higher in the presence of CHX. Values are means ± SEM of 6 experiments each.

**Figure 6 ijms-24-15464-f006:**
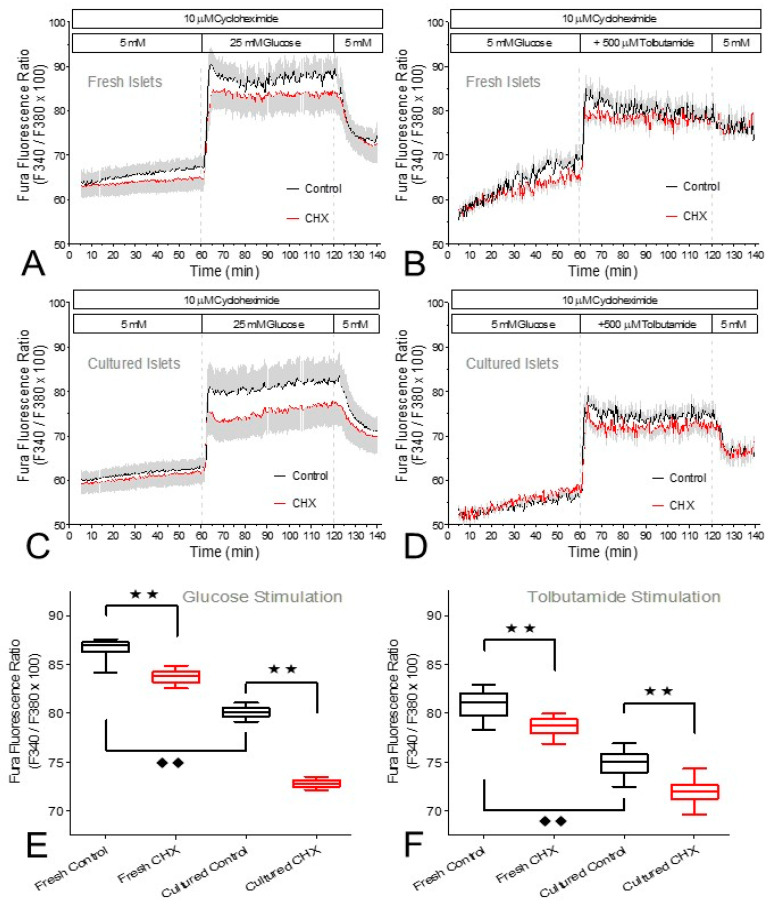
The cytosolic Ca^2+^ concentration ([Ca^2+^]_i_) during stimulation of fresh or cultured islets by glucose (**A**,**B**) or tolbutamide (**C**,**D**). Freshly isolated islets (**A**,**C**) and cultured islets (**B**,**D**) were loaded with Fura-2/AM and perifused with Krebs–Ringer medium in the presence (red traces) or absence (black traces) of 10 μM cycloheximide (CHX). After 60 min of perifusion with 5 mM glucose, the glucose concentration was raised to 25 mM or 500 μM tolbutamide was added. After a further 60 min the stimuli were washed out. For all conditions the mean values of the steady state during stimulation (70 min to 80 min) were calculated and compared (**E**,**F**). The boxes show the median, the whiskers show the 1–99 percentile of 8–12 experiments each. Asterisks denote significant differences between CHX and control (** *p* ˂ 0.01, ANOVA with Tukey’s post test), rhombus symbols denote significant differences between fresh islets and cultured islets (^♦♦^ *p* ˂ 0.01, ANOVA with Tukey’s post test).

**Figure 7 ijms-24-15464-f007:**
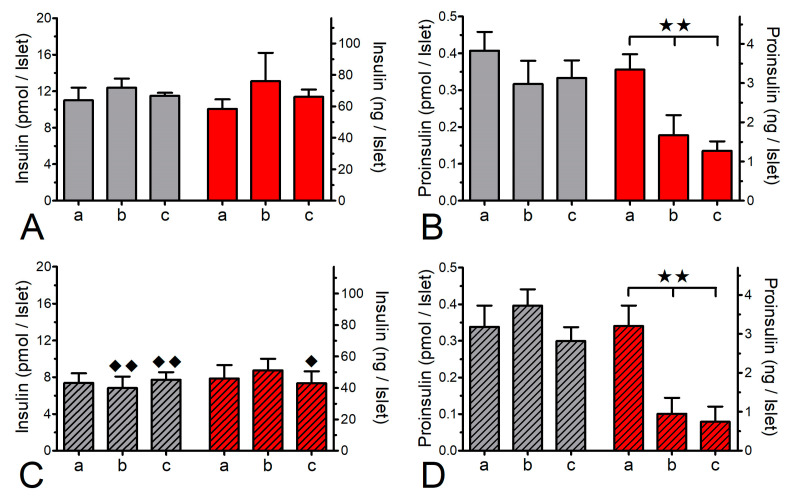
Effect of cycloheximide (CHX) on the islet contents of insulin (**A**,**C**) and proinsulin (**B**,**D**). Freshly isolated islets (open bars) or cultured islets (hatched bars) were statically incubated in the presence of 5 mM glucose for 0 min (a) or 60 min (b) and for a further 60 min in the presence of 25 mM glucose (c) in the presence (red bars) or absence (grey bars) of 10 μM CHX. After 60 min exposure to CHX the content of proinsulin, but not of insulin, was significantly reduced in fresh and in cultured islets. Note the higher content of insulin, but not proinsulin, in fresh islets. ** *p* ˂ 0.01, ANOVA with Bonferroni’s post test. ^♦^
*p* ˂ 0.05, ^♦♦^ *p* ˂ 0.01, *t*-test fresh v. cultured islets. Values are means ± SEM of 6 experiments.

**Figure 8 ijms-24-15464-f008:**
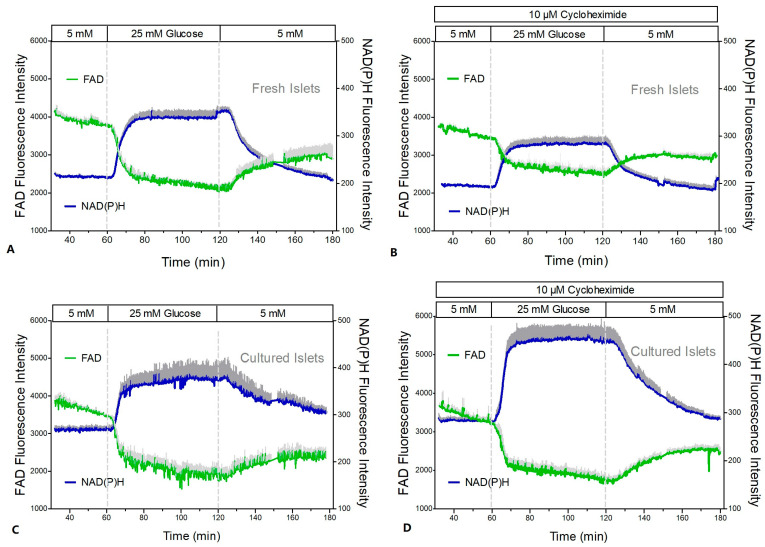
NAD(P)H- and FAD-autofluorescence during glucose stimulation of fresh and cultured islets in the presence or absence of cycloheximide (CHX). Freshly isolated islets (**A**,**B**) or cultured islets (**C**,**D**) were perifused with Krebs–Ringer medium in the absence (left graphs) or presence (right graphs) of 10 μM CHX. After perifusion with 5 mM glucose for 60 min the glucose concentration was raised to 25 mM for another 60 min and was decreased back to 5 mM thereafter. The dark gray traces denote the NAD(P)H autofluorescence, and the light gray traces denote the FAD autofluorescence. In the presence of CHX they are given as dark red or light red, respectively. Values are means ± SEM of 5 experiments each.

**Figure 9 ijms-24-15464-f009:**
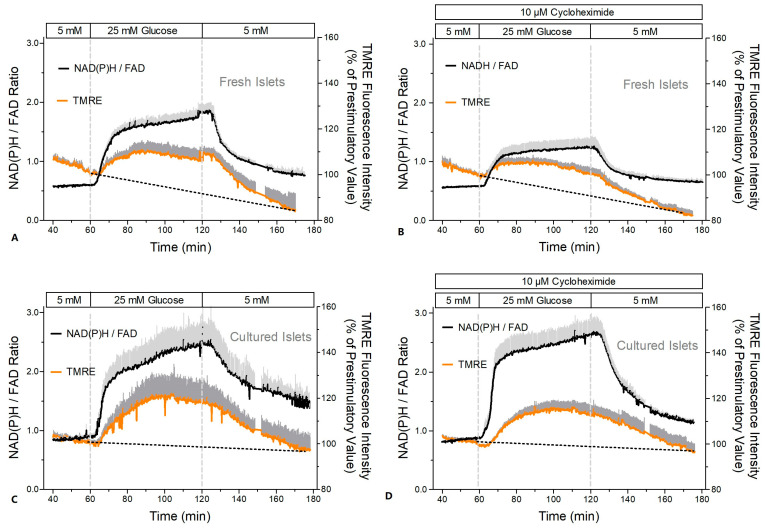
Increase in the reducing equivalents and in the mitochondrial membrane potential during glucose stimulation of fresh and cultured islets in the presence or absence of cycloheximide (CHX). Freshly isolated islets (**A**,**B**) or cultured islets (**C**,**D**) were perifused with Krebs–Ringer medium in the absence (left graphs) or presence (right graphs) of 10 μM cycloheximide (CHX, same experiments as in Figure 8). The dark gray traces denote the TMRE fluorescence as a measure of the mitochondrial membrane potentials, and the light gray traces denote the NAD(P)H/FAD ratio as a measure of the reducing equivalents. In the presence of CHX they are given as dark red or light red, respectively. Values are means ± SEM of 5 experiments each.

**Figure 10 ijms-24-15464-f010:**
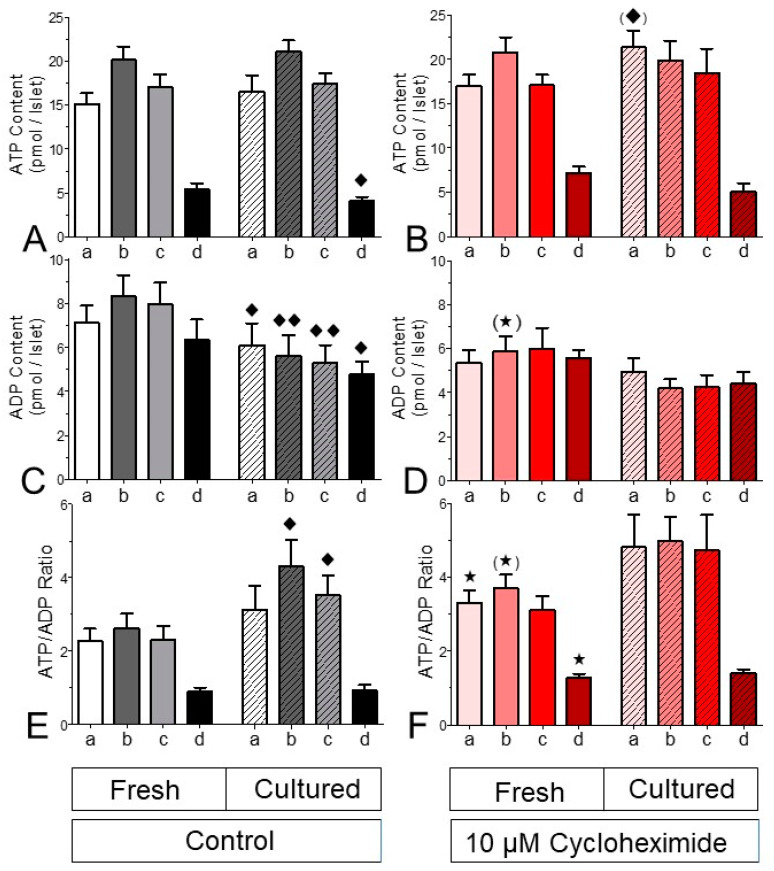
Effect of cycloheximide on the adenine nucleotide content of fresh and cultured islets. Fifteen islets each were statically incubated in parallel with the following sequence of conditions: 30 min with 5 mM glucose (a), A and an additional 20 min with 25 mM glucose (b), B and an additional 20 min with 5 mM glucose (c), C and an additional 20 min with 5 mM glucose plus 10 μM CCCP (d). For each incubation condition, the contents of ATP (**A**,**B**), ADP (**C**,**D**) and the ATP/ADP ratio (**E**,**F**) were determined. The left panels show the control values, and the right panels show the effect of 10 μM cycloheximide (CHX). Note that the significant differences in the ATP/ADP ratio are primarily caused by the decrease in the ADP contents. The presence of CHX abolished these differences. * *p* < 0.05, (*) *p* < 0.07, *t*-test, CHX-exposed v. control islets. ^♦^ *p* < 0.05, ^♦♦^ *p* < 0.01, paired *t*-test, fresh vs. cultured islets. Values are means ± SEM of 5 experiments for each condition.

## Data Availability

The data obtained in this study are available upon reasonable request.

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
