# Peer review of "Short-Term Inhibition of Translation by Cycloheximide Concurrently Affects Mitochondrial Function and Insulin Secretion in Islets from Female Mice"

_ijms, 2023, doi:10.3390/ijms242015464_

Round 1

Reviewer 1 Report

The authors treated the fresh and cultured islets with protein biosynthesis inhibitor and checked the beta cell function and metabolism and they found the mitochondrial function was affected very soon after the translation is inhibited.

However, first, there's lack of rational to to investigate protein biosynthesis inhibitors on beta cell function and they should provide more evidence to verify the inhibition of protein biosynthesis such as different kinds of inhibitors, genetic methods to inhibit protein biosynthesis.

Second, there's no clear conclusion how much  the inhibitor affect the insulin secretion and how it regulate mitochondrial function.

Third, the figures are not show clearly and the control group should always on the left compare to the treated group. And the comparison layout should always be the control and drug treated group rather than the fresh and cultured islet groups.

In conclusion, there's not much data to support the idea of the paper and the lack of a clear conclusion of the paper.

Author Response

Reviewer 1

We thank the reviewer for his/her critical comments, but would like to explain why we don´t agree with many of them.

The authors treated the fresh and cultured islets with protein biosynthesis inhibitor and checked the beta cell function and metabolism and they found the mitochondrial function was affected very soon after the translation is inhibited.

This is only part of the relevant observations. The mitochondrial function is affected as early as the insulin secretion is reduced. The reduced insulin secretion is observed only with fresh, but not with cultured islets, even though the proinsulin content was strongly reduced in both of them, suggesting that the cytosolic translation was equally affected. So, additional factors, modified by the culture period, must be involved. Our current evidence points to the mitochondrial metabolism as the critical site.

However, first, there's lack of rational to to investigate protein biosynthesis inhibitors on beta cell function and they should provide more evidence to verify the inhibition of protein biosynthesis such as different kinds of inhibitors, genetic methods to inhibit protein biosynthesis.

The rationale to investigate the effects of protein biosynthesis inhibition is described in appropriate detail in the introduction. Actually, there are two reasons to do so. First, glucose stimulation (but not depolarization-induced insulin secretion) is accompanied by stimulated insulin and insulin granule biosynthesis. It has been of interest from the very beginning of beta cell research, to clarify the role of this biosynthetic activity for the secretory response. These publications are cited (e.g. references # 15, 17 - 19). Second, in an attempt to elucidate the mechanisms of metabolic amplification, Henquin and coworkers have used cycloheximide at the same low concentration and for the same duration as used in the present investigation and have found the inhibition of translation to be ca. 90% complete (Garcia-Barrado et al., reference # 17). We have measured the insulin and proinsulin contents of fresh and cultured islets at the same time point and have found that the proinsulin content in both types of islets was strongly reduced, confirming the fast onset of action of cycloheximide. Cycloheximide is a well-characterized inhibitor of the cytosolic translation in eukaryotic cells, a review informing about its mechanism of action is mentioned in the references (reference # 20). There is no way how a genetic interference can produce a similar clear-cut inhibition within such a short time.

Second, there's no clear conclusion how much the inhibitor affect the insulin secretion and how it regulate mitochondrial function.

Frankly, we are surprised by this comment, at least by the first part of it. The description of the effect on insulin secretion is quite clear: The basal insulin secretion is not affected. The glucose-stimulated secretion of fresh islets is reduced by ca. 50% and the biphasic kinetics is abolished. The glucose-stimulated secretion of cultured islets is not diminished, and when the glucose stimulus is applied as a square wave, an enhanced second phase occurs. As for the mechanism of how cycloheximide affects mitochondrial function we have mentioned the well-known fact that except for 13 proteins of the respiratory chain all other mitochondrial proteins are generated in the cytosol and imported into the mitochondria. So, even though cycloheximide acts specifically on eukaryotic translation, the consequences of its inhibitory action are likely pleiotropic from the beginning on and will broaden with time, depending on the half-time of the mitochondrial proteins which need to be replaced.

To support our contention that in beta cells mitochondrial proteins must be replaced within a short time frame, we have performed experiments with inhibitors of prokaryotic//mitochondrial translation. Preliminary data show that this has similar consequences for the stimulated secretion, this observation is now included in the discussion section (p. 15, 1st para, 2nd last sentence).

Third, the figures are not show clearly and the control group should always on the left compare to the treated group. And the comparison layout should always be the control and drug treated group rather than the fresh and cultured islet groups.

From the beginning of the project, the experimental design included two independent variables: inhibition of translation and effect of islet culture. So, the results are best presented as a square lattice. But this is not always possible, depending on the complexity of the graphical presentation (see e.g. Figs 1 and 2). Additionally, we have in the initial part the comparison between glucose stimulation and depolarization-induced secretion by tolbutamide, which renders the straightforward arrangement as suggested by the reviewer impracticable.

In conclusion, there's not much data to support the idea of the paper and the lack of a clear conclusion of the paper.

We hope that the above arguments convince the reviewer that inhibition of cytosolic translation affects insulin secretion not by reducing granule formation or some off-target effects (the effect should be the same in fresh and cultured islets), but by indirect interference with mitochondrial signal generation.

Reviewer 2 Report

the paper examines the role of the pancreatic beta cell in sensing the availability of nutrients, particularly glucose, in the blood and translating it into insulin secretion rates. They investigated the potential role of protein synthesis and its relationship to insulin secretion, particularly under the influence of cycloheximide, a known inhibitor of eukaryotic translation. The study was performed on both freshly isolated and cultured islets. The authors concluded that beta-cell mitochondrial function is impaired shortly after translation is inhibited, which is closely related to the inhibition of secretion.

The authors have analyzed the phenotypes of the phenomenon in various experiments, supported by designed experimental setups to explore shifts in mitochondrial function. Nevertheless, before publication, certain points merit correction:

- The paper introduces terms such as "cataplerosis" and "metabolic amplification" without adequate context. A brief definition or explanation of these terms within the manuscript might improve its readability.

- In Figure 3E, although glucose-induced OCR in cultured islets appears to be invariant after CHX treatment, the graph showed significance difference. please check raw data.

- Figure 6, panels A, C, and D show a decrease in the Fura ratio after the 120-minute mark, while panel B appears to be consistent. Please explain it.

- I recommend to write 'time ' in the x-axis in Figure 7

- Figures 9A and C are in grayscale. please change it.

-Consider moving the labels identifying each panel (e.g., A, B, C,...) to the upper left corner of each figure for uniformity and ease of reference.

Author Response

Reviewer 2

The paper examines the role of the pancreatic beta cell in sensing the availability of nutrients, particularly glucose, in the blood and translating it into insulin secretion rates. They investigated the potential role of protein synthesis and its relationship to insulin secretion, particularly under the influence of cycloheximide, a known inhibitor of eukaryotic translation. The study was performed on both freshly isolated and cultured islets. The authors concluded that beta-cell mitochondrial function is impaired shortly after translation is inhibited, which is closely related to the inhibition of secretion.

The authors have analyzed the phenotypes of the phenomenon in various experiments, supported by designed experimental setups to explore shifts in mitochondrial function. Nevertheless, before publication, certain points merit correction:

We thank the reviewer for his/her careful reading and constructive criticism.

- The paper introduces terms such as "cataplerosis" and "metabolic amplification" without adequate context. A brief definition or explanation of these terms within the manuscript might improve its readability.

In the introduction we first explain the terminology of “triggering pathway” and “amplifying pathway” as suggested by Henquin and then go on to explain what is meant by “metabolic amplification”: To distinguish these mitochondrially derived signalling from the receptor-mediated enhancement of insulin secretion, it is termed “metabolic amplification”. We then go on to explain the probable mechanism: “Metabolic amplification likely involves cataplerosis, i.e. the export of citric acid cycle metabolites [12,13], which places it into competition with another essential function of nutrient secretagogues, namely the stimulation of protein synthesis.”

Furthermore, we have introduced an additional sentence in the discussion to emphasize the central role of the mitochondria in the glucose sensing of the beta cell and have included a new review article on this role in the references (new reference # 36).

- In Figure 3E, although glucose-induced OCR in cultured islets appears to be invariant after CHX treatment, the graph showed significance difference. please check raw data.

The reviewer probably refers to the marginally significant difference between the OCR increase of CHX-exposed and control islets, both freshly isolated. This is part of the statistical evaluation of the OCR registrations depicted in Fig. 1A. In Fig. 1A the control OCR increases from a moderately lower level than the CHX OCR, but during stimulatory glucose both OCR are on the same level. In Fig. 3E the difference between the level prior to and the level during glucose stimulation is considered. Also, the statistical testing was done by paired comparison. This can reveal statistically significant differences even when the mean values appear similar due to data scattering.

In case the reviewer´s comment refers to the symbol above the bar depicting the cultured control islet in Fig. 3E, this is how the legend reads: “the rhombus symbol denotes a significant difference between fresh islets and cultured islets”. So, the OCR increase is smaller with cultured islets than with fresh islet, which fits to the visual inspection of the respective curves in Fig. 1A and Fig. 1B. As a general rule, asterisks denote significant differences between CHX-exposed islets and control, rhombus symbols denote significant differences between fresh and cultured islets. To conclude, we are confident that no error has occurred in the data processing.

- Figure 6, panels A, C, and D show a decrease in the Fura ratio after the 120-minute mark, while panel B appears to be consistent. Please explain it.

The reviewer is correct in that the wash-out of tolbutamide should lead to a visible decrease of the cytosolic Ca2+ concentration. With cultured islets this is clearly happening (Fig. 6D), whereas the decrease is very small with fresh islets (Fig. 6B). We have repeatedly seen that at basal glucose the Fura ratio is higher in fresh islets than in cultured islets. Most likely this really means that the cytosolic Ca2+ is higher. This puts a strain on the cellular ATP generation, since the Ca2+ has to be pumped out or sequestered by ATP-consuming transport mechanisms. The ATP generation by high glucose may be the reason why the Ca2+ decrease is more efficient after glucose (Fig. 6A) than after tolbutamide (Fig. 6B). 

- I recommend to write 'time ' in the x-axis in Figure 7

The reason for us to mention the incubation time in the figure legend is that the data were not derived from a continuous perifusion, but from separate static incubations, mimicking the course of the perifusion protocol. The same is true for the labeling in Fig. 10.

- Figures 9A and C are in grayscale. please change it.

This coloring was chosen because red symbolizes CHX-exposure and black symbolizes the respective control. However, since the traces of CHX-exposure and of control are in separate panels, a different coloring scheme can be used without causing confusion. So, the traces of NAD(P)H, FAD, and TMRE are now given in the color of their fluorescence emission, the NAD(P)H/FAD-ratio is given in black.

Consider moving the labels identifying each panel (e.g., A, B, C,...) to the upper left corner of each figure for uniformity and ease of reference.

We have tried this change of labelling, but have finally come to the conclusion that it does not provide an easier orientation. Since all labelling of the panels is located at the lower left side, it should be sufficiently uniform as not to confuse the viewer.

Round 2

Reviewer 1 Report

The response from the authors are acceptable but still consider it may lack of significant content.